Brain virtual histology and volume measurement of a lizard species (Podarcis bocagei) using X-ray micro-tomography and deep-learning segmentation

Zhou Tunhe 1 tunhe.zhou@su.se
Dragunova Yulia 1 2 3
http://orcid.org/0000-0001-5592-8963 Triki Zegni 4 5
1 SUBIC, Stockholm University , Stockholm , Sweden
2 KTH Royal Institute of Technology , Stockholm , Sweden
3 Nuclear Medicine and Medical Physics, Karolinska University Hospital , Stockholm , Sweden
4 Department of Zoology, Stockholm University , Stockholm , Sweden
5 Institute of Biology, University of Neuchâtell , Neuchâtel , Switzerland
Brygadyrenko Viktor
Electronic publication date: 2025 Sep 1
Publication date: 2025
Volume: 13
Electronic Location ID: e19672
Received 2024 Oct 16; Accepted 2025 Jun 6
Copyright: © 2025 Zhou et al.
Copyright year: 2025
Copyright holder: Zhou et al.
License: This is an open access article distributed under the terms of the Creative Commons Attribution License, which permits unrestricted use, distribution, reproduction and adaptation in any medium and for any purpose provided that it is properly attributed. For attribution, the original author(s), title, publication source (PeerJ) and either DOI or URL of the article must be cited.
License URL: https://creativecommons.org/licenses/by/4.0/

Keywords: Lizard, Brain, X-ray, Segmentation

Funding: Swiss National Science Foundation P400PB_199286, PZ00P3_209020 Association for the Study of Animal Behaviour (ASAB) Stockholm University Brain Imaging Centre SU FV-5.1.2-1035-15 This work was financially supported by the Swiss National Science Foundation (Nos. P400PB_199286 and PZ00P3_209020 to Zegni Triki) and a research grant from The Association for the Study of Animal Behaviour (ASAB) to Zegni Triki. Brain data acquisition was supported by a grant to the Stockholm University Brain Imaging Centre (SU FV-5.1.2-1035-15). The funders had no role in study design, data collection and analysis, decision to publish, or preparation of the manuscript.

==============================
There is an increasing emphasis on understanding individual variation in brain structure—such as overall brain size and the size of specific regions—and linking this variation to behaviour, cognition, and the driving social and environmental factors. However, logistical challenges arise when studying the brain, especially in research involving wild animals, such as dealing with small sample sizes and time-consuming methods. In this study, we used wild lizards, Podarcis bocagei, as our model. We developed an efficient and accurate method that combines X-ray micro-tomography and deep-learning segmentation to estimate the volume of six main brain areas: the olfactory bulbs, telencephalon, diencephalon, midbrain, cerebellum, and brain stem. Through quantitative comparisons, we show that a sufficiently trained deep-learning neural network can be developed with as few as five samples. Using this trained model, we obtained volume data for the six brain regions from 29 brain samples of Podarcis bocagei. This approach drastically reduced the time needed for manual segmentation from several months to just a few weeks. We present a comprehensive protocol detailing our methods, which includes sample preparation, X-ray tomography, and 3D volumetric segmentation. This work collectively provides valuable resources that can assist researchers not only in animal behaviour and physiology, but also in biomedical research and computer sciences.

Introduction

There has been an increasing interest in studying the individual variation in animal behaviour and cognition and understanding the underlying neural correlates (Gonda, Herczeg & Merilä, 2013; Thornton & Lukas, 2012; Triki & Bshary, 2022; Triki et al., 2019). Understanding intraspecific variation in the brain and linking it to the individual-level ecology, like social group size, dominance hierarchy, foraging behaviour, mating strategies, and predator avoidance (Guadagno & Triki, 2023; Kolm et al., 2009; Triki et al., 2020, 2019; Vega-Trejo et al., 2022), among other selective pressures, can help us to understand better brain development and, ultimately, brain evolution.

Ectotherms, including fish, amphibians, and reptiles, represent a promising vertebrate group for exploring the relationship between ecology, cognition, behaviour, and the brain. Their ability to maintain neurogenesis in adulthood and generate neural tissue when necessary for adaptation enables the detection of variation within species and even within populations (Gonda, Herczeg & Merilä, 2013; Zupanc, 2006). However, several challenges emerge when studying vertebrate brain morphology and its relationship to ecology (social and environmental conditions). For example, to separate species-specific effects and account for phylogenetic signals, comparative brain studies require extensive sample sizes and a diverse array of species. This is difficult due to the labour-intensive methods needed to collect brain morphology data, which often results in studies with small sample sizes. Additionally, in comparative research, it is essential to standardise methodologies and reduce observer bias (Tuyttens et al., 2014).

A plausible solution involves measuring total brain size, which seems relatively straightforward, as it necessitates fewer complex techniques to ascertain volume or weight. Additionally, it is recognised as one of the most extensively studied traits in evolutionary biology, both in existing and extinct species (Isler et al., 2008). Acquiring the size of different brain regions complicates the matter. In neuroecology, cost-effective techniques such as ellipsoid measurements are frequently used to estimate the sizes of brain regions in several teleost fish species from lateral, ventral, and dorsal brain pictures, e.g., cichlids, guppies, and gobies (Pollen et al., 2007; Triki et al., 2023; White & Brown, 2015). The technique uses pictures of brains from lateral, ventral, and dorsal sides and estimates the volume of each brain lobe from its length, width, and height, fitting these dimensions into the ellipsoid formula: volume = (length × width × height) × π/6. While the method offers useful insights, it has low accuracy by simply assuming that all the lobes are ellipsoidal in shape (White & Brown, 2015). Consequently, the ellipsoid method is not generalizable across species, particularly when brain shapes differ markedly from an elliptical form. Ectothermic species typically have long, narrow, and tubular brains, with a more linear organization in which different regions are arranged sequentially. Their brain lobes are often more ellipsoidal in shape, which may align more closely with the assumptions of the ellipsoid method. In contrast, endothermic brains tend to be more compact and rounded, making the method less suitable for these species (Striedter, 2005).

Another method widely used for estimating brain region sizes is the histological technique. It involves sectioning the brain into slices, then staining and estimating the surface area of each region on a brain slice, summing these areas, and multiplying them by the thickness of each slice (White & Brown, 2015). The histological technique estimates various lobe volumes more precisely than the ellipsoid method. Additionally, it considers the irregular shapes of lobes and the beginning and end of ventricles, aspects that the ellipsoid estimation cannot account for. Nonetheless, the histological method has significant limitations, mainly because obtaining brain region measurements is extremely time-consuming and tissue-destructive (Cnudde et al., 2008; Johnson et al., 2006; Kuan et al., 2020), rendering it less appealing than the ellipsoid technique.

In the past decade, modern imaging methods, previously reserved for medical usage and clinical research, have made a significant impact on neuroecology, such as magnetic resonance imaging (MRI) and X-ray tomography. These methods provide 3D images of the internal structures in a non-destructive way, allowing accurate volumetric measurement, as well as repeatability and sample reusability, which is especially important for wild specimens and animal welfare purposes.

So far, two studies have provided 3D models of lizard brains, the tawny dragon (Ctenophorus decresii) and the Bearded Dragon (Pogona vitticeps), using MRI (Foss et al., 2022; Hoops et al., 2018, 2021). In this study, we provide a detailed step-by-step protocol (Zhou, Dragunova & Triki, 2024) for extracting 3D brain morphometric data using X-ray micro-tomography (microCT) from a common lizard, Podarcis bocagei. To increase efficiency and reduce time-consuming sample preparation, we prepared whole heads instead of dissecting the brains. This also avoids potential damage during dissection, embedding and sectioning as in traditional histological methods.

Using microCT 3D data and deep-learning neural network (Lösel et al., 2020; Ronneberger, Fischer & Brox, 2015), we anatomically segmented six distinct major brain regions: olfactory bulbs, telencephalon, diencephalon, midbrain, cerebellum, and brain stem (Bruce, 2009) of 29 specimens. Deep learning segmentation for 3D images is fast growing in medical imaging (Liu et al., 2021), where the training data size is generally in the range of hundreds (Oktay et al., 2018). It is uncommon to have such large datasets for studies on wild animals. However, our study demonstrated that deep learning segmentation could yield satisfactory results with as few as five training sets (brains), which has the potential to significantly reduce manual segmentation efforts for both large and small-scale studies. Portions of this text were previously published as part of a preprint (Zhou, Dragunova & Triki, 2024).

Materials and Methods

Animals

This study was approved under ethics permit No. 873-876/2021/CAPT granted by Instituto de Conservação da Natureza e das Florestas (ICNF), Portugal. Animal capture, handling and euthanasia were conducted according to the regulations of the University of Porto.

Our samples comprised 29 male lizards (Podarcis bocagei) with a body mass of 4.38 ± 0.70 g (mean ± SD). They were captured in May 2021 around the Research Center in Biodiversity & Genetic Resources (CIBIO) campus at the University of Porto in Portugal. The animals were first used in another study to test their behaviour and cognitive abilities (unpublished data). Upon finishing the tests, all animals were euthanised with an anaesthetic intramuscular injection of Zoletil (10 mg/kg), followed by an intraperitoneal injection of sodium pentobarbital (80 mg/kg). The whole heads of the lizards were placed in a fixative solution composed of 4% paraformaldehyde in phosphate-buffered saline (PBS) for 10 days before being rinsed and stored in PBS. Then, the samples were shipped to Stockholm University for brain morphology analyses.

Sample preparation

The lizard head samples were stained with phosphotungstic acid (PTA) to enhance the tissue contrast for X-ray imaging (Fig. S1). The staining protocol was adapted from Lesciotto et al. (2020) to our samples for optimised scans. It started with the dehydration process followed by staining. In the dehydration process, we adhered to the following steps:

Step 1: placing the samples in 30% ethanol in PBS for 1 day;

Step 2: placing the samples in 50% ethanol in PBS for 1 day;

Step 3: placing the samples in 70% ethanol in PBS for 1 day;

Step 4: placing the samples in a solution with a ratio of 4:4:3 volumes of ethanol, methanol, and water for 1 h;

Step 5: placing the samples in 80% methanol in ultrapure water for 1 h;

Step 6: placing the samples in 90% methanol in ultrapure water for 1 h.

After that, we proceeded with the PTA staining. The samples were immersed in 0.7% PTA in 90% methanol in ultrapure water. During the staining period, we consistently assessed the staining quality by checking tissue contrast with X-ray scans every week. Not all samples reached optimal staining simultaneously; rather, samples with relatively larger heads required prolonged staining times. Overall, the samples needed between 23 and 30 days of staining to reach optimality. We defined optimal staining as the point at which soft tissue contrast across major brain regions was clearly visible and no longer improved with additional time in the staining solution.

X-ray microCT scan

The samples were scanned using Zeiss Xradia Versa 520 at Stockholm University Brain Imaging Centre, with the samples being stabilised inside 5 mL plastic tubes and remained in the staining solution, as shown in Fig. 1. The X-ray source was set to have a voltage of 100 kV and a power of 9 W. The 0.4x objective and a charge-coupled device (CCD) camera were coupled with a scintillator. The effective voxel size was 17.4 μm, with the optical and geometrical magnification being compensated. The scan consisted of 801 projections over 212 degrees with 1 s exposure time for each projection. In total, one scan took 36 min, including reference images and readout time of the CCD camera. Autoloader was utilised to change samples automatically to minimise the manual work. The tomography reconstruction was done automatically with Zeiss Scout-and-Scan software right after the scans, and the output was 16-bit grey value tiff image stacks.

Figure 1 Experiment setup showing the X-ray scanner equipment and the samples prepared in a queue to be scanned automatically.

Semi-automatic segmentation of training datasets

It was not possible to manually place all samples in exactly the same position with an accuracy of tens of microns. For this reason, the first step in preparing the data were to align and crop all the images, as demonstrated in Fig. 2A. This was performed using the software Dragonfly. This step may be overlooked but it was crucial for a consistent brain volume measurement, especially the brainstem, and later for efficient neural network training.

Figure 2 (A) The images are aligned and cropped to keep the brain for fast segmentation. (B and C) Demonstration of manual segmentation. (D) A flowchart of the segmentation procedure.

The training dataset was segmented utilising semi-automated methods and was comprised of three distinct steps. The first step was manual segmentation of approximately 20–30 slices in total of the three planes in Dragonfly (Figs. 2B, 2C). The second step was random walk interpolation in Biomedisa (Lösel & Heuveline, 2016; Lösel et al., 2020). The third and final step was checking the segmentation and correcting manually any inconsistencies if needed. As the boundaries of certain brain regions can be indistinct in some areas, we have manually delineated these brain regions in the initial step of the segmentation, following brain anatomy descriptions and brain atlases from lizard species (Bruce, 2009; Hoops et al., 2018; Menezes Freitas, Paranaiba & Lima, 2023). A sketch of the procedure from semi-automatic labelling to deep-learning segmentation is shown in Fig. 2D.

Deep-learning based segmentation

We tested and compared two deep learning algorithms, namely Biomedisa (Lösel et al., 2020) and AIMOS (Schoppe et al., 2020). They have been recognised as two of the most user-friendly open-source software options, which constitutes a crucial factor for researchers in various fields to effectively employ the method, facilitated by straightforward installation and a relatively easy learning curve. Both algorithms are based on U-Net (Ronneberger, Fischer & Brox, 2015), which is a type of convolutional neural network (CNN) named for its unique U-shaped architecture, with an autoencoder with skip connections. Compared to other CNN models, autoencoders delineate clear boundaries while maintaining a simpler model architecture. Additionally, the decoder in such networks is adept at forming distinct boundaries from the extracted features. However, a significant challenge in using autoencoders is the potential oversimplification of images during the encoding phase (Ghosh et al., 2019). To overcome this, linear skip connections are employed extensively. These connections enhance the accuracy of the segmentation maps by merging both elementary and complex features from different layers of the U-Net. Specifically, in U-Net, skip connections directly transfer the detailed activation outputs from the encoder to their corresponding layers in the decoder, facilitating precise feature mapping.

A. Biomedisa

We used the Biomedisa online application. The brain images and labels are compressed in two TAR files. Biomedisa follows 3D U-Net (Çiçek et al., 2016) and uses Keras with TensorFlow as framework (Abadi et al., 2016). Before training the network, images were automatically scaled and normalized by Biomedisa, using the parameters shown in Table 1. The training time for using different numbers of datasets is listed in Table 2. The rest of the dataset was predicted using the trained neural network where each took less than 1 min.

Table 1 Neural network training parameters Biomedisa.

Network architecture	32-64-128-256-512-1024	
Number of epochs	200	
Batch size	24	
Stride size	32	
Image scale	256 * 256 * 256	

Table 2 Training time of different dataset numbers using Biomedisa.

Number of training sets	Time (min)	
1	55	
3	150	
5	248	
7	341	
9	430	
11	540	

B. AIMOS

We used another user-friendly software, AIMOS (Schoppe et al., 2020), for comparison. AIMOS uses Python on a local computer and PyTorch as its framework (Paszke et al., 2019). The software employs an architecture similar to U-Net with six levels of encoding and decoding blocks. The parameters for training are reported in Table 3. AIMOS provides pre-trained networks available for microCT scans making the training time potentially shorter. In AIMOS, the data was split into three sets: a training set for model weight optimisation, a validation set for hyper-parameter optimization, and a test set for evaluation. The pipeline requires at least two extra datasets, compared to Biomedisa, where the validation ratio could be set to 0.

Table 3 The default neural network training parameters in AIMOS.

Network architecture	32-64-128-256-512-768	
Number of epochs	30	
Batch size	32	
Image scale	256 * 256	

Evaluation metrics

The dice score or dice similarity coefficient for each brain region was calculated using Eq. (1),

(1) DSC=2×|T∩P||T|+|P|,

where T is the set of voxels outlined by the manually segmented ground truth, and P is the set of voxels outlined by the predicted segmentation (Bertels et al., 2019). The dice score is between 0 and 1, where 1 indicates a perfect prediction. The dice score was calculated for each predicted brain region in 3D separately and then averaged to obtain the dice score for the whole brain.

Since our primary objective is to estimate brain region volumes, it is important to assess the accuracy of the segmentation predictions generated by the deep learning algorithm. This can be achieved by calculating the average relative error of the predicted region volumes (Eq. (2)),

(2) R.E.=|Vp−VT|VT×100%,

where VP is the volume of the prediction, and VT is the ground-truth volume. The volumes of the segmented brain regions are calculated from the number of voxels of each label multiplied by the voxel size. The relative measurement error was calculated separately for all the brain regions in one sample and then averaged to obtain the relative error for each brain.

Results

Virtual histology and 3D model from X-ray micro-tomography (microCT)

Figures 3 and 4 display examples of virtual histology sections from a single brain in coronal, sagittal, and horizontal orientations. Several brain areas are labelled with references from Hoops et al. (2018), Menezes Freitas, Paranaiba & Lima (2023), Naumann et al. (2015) (abbreviations in Table 4). The brain virtual sections in Figs. 3 and 4 were generated with a mask from segmentation without manual brain dissection. The results also provide virtual histology of the whole head, offering detailed information about tissues beyond the brain. For example, in Fig. 5A, the olfactory nerves can be identified, and the diameter is approximately 26 μm as shown in the inset.

Figure 3 Coronal slices at positions indicated in the upper left subfigure.

The abbreviations of the brain parts are listed in Table 4.

Figure 4 Examples of (A) horizontal and (B) sagittal slices from virtual histology at positions indicated in the upper subfigure.

The abbreviations of the brain parts are listed in Table 4.

Table 4 Abbreviations of brain regions.

Abbreviation	Brain region	
4V	Fourth ventricle	
AOB	Accessory olfactory bulb	
aon	Accessory olfactory nerve	
Ce	Cerebellum	
DC	Dorsal cortex	
DMC	Dorsomedial cortex	
DVR	Dorsal ventricle ridge	
GL	Granular layer of the cerebellum	
LC	Lateral cortex	
lfb	Lateral forebrain bundle	
LV	Lateral ventricle	
MC	Medial cortex	
ML	Molecular layer of the cerebellum	
MOB	Main olfactory bulb	
on	Olfactory nerve	
OT	Optic tectum	
ot	Optic tract	
TV	Tectal ventricle	

Figure 5 Virtual histology of the head of one Podarcis bocagei.

(A) Zoom in of a sagittal section on the olfactory system, showing the olfactory nerves (on) and accessory olfactory nerves (aon). (B) A horizontal section showing how the optic nerve connects the brain and the retina, as well as the cornea and iris of the eyes. (C and D) Transversal and sagittal cross sections of the head showing structures such as the teeth, salivary gland, the organ of Jacobson in the olfaction system.

Brain region segmentation

Using deep learning methods, we segmented six main brain regions from 29 specimens, namely, the olfactory bulb, telencephalon, diencephalon, midbrain, cerebellum, and brainstem (see Fig. 6) (also see Video S1). The average volume values of all 29 segmented brains are reported in Fig. 7 using MATLAB, with the standard deviation as the error bars (see Table S1).

Figure 6 3D rendering of the X-ray microCT images showing the external and internal of the head, and the main parts of the brain.

The brain regions are: the olfactory bulb (green), telencephalon (purple), diencephalon (red), midbrain (blue), cerebellum (yellow), and brainstem (pink).

Figure 7 The volumes of the brain regions.

The volumes of the six brain regions of the 29 specimens were measured from the segmentation. The error bars show the standard deviation, and the bars show the mean value of the volumes. The segmentation was done using deep-learning neural network, except the ones used in the training datasets, that were segmented semi-automatically.

The results of using both deep learning methods, Biomedisa and AIMOS, are compared and presented in Fig. 8 (Table S2). We found that, because AIMOS is a 2D approach, the orientation of the slices influenced the results (Fig. S2). We identified that the sagittal plane works best for AIMOS and therefore chose it for the comparison with Biomedisa. In contrast, Biomedisa operates on a 3D representation and is not restricted to a specific data plane orientation. In Fig. 8, five separate networks were trained for each algorithm using three, five, seven, nine, and 11 images, where each trained neural network was then used to predict four test image stacks. The predicted segmentations were evaluated using the average dice score (DSC) (Bertels et al., 2019) and the average relative error of brain region volumes (RE). It can be seen that as few as three samples provided sufficient accuracy with DSC > 0.9 using Biomedisa. As AIMOS required two samples from the labelled data for validation, the input data size cannot be smaller than three. For both AIMOS and Biomedisa, RE was about 4–5% when the training sets were 5 or more, as shown in Fig. 8 (Table S2).

Figure 8 Evaluation of the deep-learning-based segmentation.

The values of the plots are listed in the Supplemental File.

Overall, comparing the two algorithms revealed a visible difference for small data sizes. At an input size of 3, Biomedisa far outperforms AIMOS. Nevertheless, when the input data size exceeds 5, this difference in performance disappears.

Discussion

Twenty-nine specimens of common lizard Podarcis bocagei were scanned using X-ray microCT to generate 3D images with internal structures in high resolution. Six brain regions, olfactory bulbs, telencephalon, diencephalon, midbrain, cerebellum, and brain stem, were segmented, and their volumes were measured. Two deep learning segmentation algorithms were applied and compared quantitatively. The results have shown that as few as five training datasets (five samples) were sufficient for both algorithms. Biomedisa operates in 3D and is not restricted to a distinct plane orientation. AIMOS uses 2D U-Net, the orientation of the slices influenced the results. For AIMOS, the sagittal plane works best in this study as sagittal slices contain most of the brain regions simultaneously.

The methods described in this study offer an efficient protocol for achieving 3D imaging acquisition and volume measurement with minimal manual work and high accuracy in several aspects. First, it eliminates manual work in brain dissection and sectioning like in classical histological techniques. Second, it offers more accurate volumetric measurements compared to the ellipsoid method (White & Brown, 2015). Third, it significantly reduces manual labelling time by leveraging smart interpolation and deep learning prediction. Instead of manually segmenting approximately 200 slices from 29 samples, we trained the model using just 30 slices from five samples—achieving a 39-fold speedup. This translates to 2 weeks of work with our protocol compared to 78 weeks (18 months) of manual labelling. Finally, it increases data accessibility by facilitating the reuse and reproducibility of the findings.

In neuroecology, there are several limitations to conducting research that links brains to behaviour and cognition, especially in wild animals. Studying wild animals and obtaining sufficiently large sample sizes can prove challenging (Bshary & Triki, 2022). To address this, researchers often tend to maximise the amount of data collected from wild specimens, both to compensate for small sample sizes and to prioritise animal welfare (Bee et al., 2020). Therefore, creating a database with 3D brain scans beyond the scope of individual studies would be highly beneficial, as it would enable other researchers to access these data and explore new questions related to different traits. For instance, in addition to measuring the size of brain regions, high-resolution 3D scans can provide more comprehensive data on other traits, such as those associated with vision (e.g., Fig. 5B). Compared to traditional histological techniques, these scans can offer valuable insights without the risk of deformation during dissection. Accurate 3D modelling is critical for simulating the optical properties and visual systems in various species, including humans, by extracting the 3D morphology of eye structures (MercuȚ et al., 2020; Taylor et al., 2020). This enables precise estimation of key parameters such as resolution, focal length, and field of view. Moreover, other anatomical information, such as tooth development, skull structure, and jaw anatomy, can also be gathered from these scans (see examples in Figs. 5C, 5D) (Ballell et al., 2024; Beazley et al., 1998; Urošević, Ljubisavljević & Ivanović, 2014; Zahradnicek, Horacek & Tucker, 2012).

While X-ray microCT may not match the 2D resolution of histological images, it offers isotropic resolution in three dimensions, providing more detailed information in the sectioning dimension and enabling virtual histology in any direction from the same scan. This reduces the need for manual sample sectioning and minimises the number of animals required, making X-ray microCT an ideal tool for volumetric studies. MicroCT is not as cost-efficient as ellipsoidal volume estimation, but it is less costly than MRI. Additionally, compared to MRI, X-ray microCT provides higher resolution and better contrast for bones, making the data potentially useful for other studies. The staining protocol in this study can be adjusted according to the requirements of the studies, for example, iodine staining can be used for faster staining, but it is not suitable for long-term storage due to the destaining (Callahan et al., 2021). In contrast, for museum specimens or other samples that need to be destained afterwards, extra steps, including NaOH washing, are required if PTA is used (Hanly et al., 2023).

Importantly, the segmentation protocol outlined here is not limited to X-ray microCT data; it can be applied to any 3D scans with different training datasets tailored to various imaging modalities. Additionally, we have made our 3D microCT data, along with the corresponding labels and trained neural network, freely available (Zhou, Dragunova & Triki, 2025).

With the growing number of studies in computer science highlighting the potential of deep learning for various image processing tasks, this study aims to provide a practical and easy-to-follow guideline. We hope this protocol will be valuable to researchers across diverse fields beyond ecology and evolution, enabling faster, more reliable, and reproducible brain segmentation. Computer scientists could also benefit from accessing our data to develop more efficient algorithms for various tasks, as they are currently working on constructing algorithms to “segment anything” for general 2D images (Kirillov et al., 2023), as well as for 3D medical imaging (Ma et al., 2024). Without a sufficient available data bank, the development of algorithms for animals other than standard model organisms would not be possible. Future work can focus more on improving the training efficiency, combined with data augmentation and simulation, to push the limits of training dataset size further and bring manual annotation efforts to a minimum.

Supplemental Information

Supplemental Information 1 3D rendering of one scan and the segmented brain regions.

Supplemental Information 2 Raw data used to generate the values used in Figure 7.

The volumes were measured from segmentations using deep-learning network and semi-automatic interpolation.

Supplemental Information 3 The dice score and relative error for Figure 8 for the two deep learning methods Biomedisa and AIMOS.

Supplemental Information 4 Example of sample staining procedure.

(a) An X-ray projection image of a sample, with the arrow indicating an area where the brain and eye were not fully stained. (b) A slice from the microCT reconstruction showing that the brain appears “missing” due to incomplete staining. (c, d) Examples of a well-stained sample for comparison.

Supplemental Information 5 One sample labelled by predictions from AIMOS trained with data in coronal (yellow) and sagittal (red) planes.

We are grateful to Lekshmi Sreelatha, Zbyszek Boratynski, Miguel Carretero (and his team), Bárbara Bastos, and Philipp Lehmann for generously providing us with the lizard samples and for their efforts in organising the logistics of animal capture, handling, sampling, and shipping. We also thank Anna Burvall for her advice on the segmentation task. ChatGPT was used to correct some grammatical issues during proofreading.

Additional Information and Declarations

Competing Interests

Zegni Triki is an Academic Editor for PeerJ.

Author Contributions

Tunhe Zhou conceived and designed the experiments, performed the experiments, prepared figures and/or tables, authored or reviewed drafts of the article, and approved the final draft.

Yulia Dragunova performed the experiments, analyzed the data, prepared figures and/or tables, authored or reviewed drafts of the article, and approved the final draft.

Zegni Triki conceived and designed the experiments, performed the experiments, authored or reviewed drafts of the article, and approved the final draft.

Animal Ethics

The following information was supplied relating to ethical approvals (i.e., approving body and any reference numbers):

Ethics permit was granted by Instituto de Conservação da Natureza e das Florestas (ICNF), Portugal (No. 873-876/2021/CAPT).

Data Availability

The following information was supplied regarding data availability:

The 3D microCT data of the lizard head used in Figs. 5 and 6 is available at FigShare:

- Zhou, Tunhe; Dragunova, Yulia; Triki, Zegni (2024). X-ray microCT lizard brain data and labels. Trained network using Biomedisa with different number of training datasets. Stockholm University. Dataset. https://doi.org/10.17045/sthlmuni.26164570.v3.

The raw data for Fig. 7 is available in the Supplemental File.

The segmented labels for measuring the brain volumes in Fig. 7 and some trained networks are available at Figshare: Zhou, Tunhe; Dragunova, Yulia; Triki, Zegni (2024). X-ray microCT lizard brain data and labels. Trained network using Biomedisa with different number of training datasets. Stockholm University. Dataset. https://doi.org/10.17045/sthlmuni.26164570.v3.

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
