# Peer review of "Brain virtual histology and volume measurement of a lizard species (Podarcis bocagei) using X-ray micro-tomography and deep-learning segmentation"

_PeerJ, doi:10.7717/peerj.19672_

## Round 0.1 · original submission · Major Revisions

Dear authors, unfortunately, all reviewers note the low scientific and literary level of your manuscript. I may still have to reject it from publication after you make corrections and additions. However, I still hope that you will be able to use the reviewers' recommendations and send a radically improved version of this article. I hope that you will be able to involve highly qualified specialists in editing the text and data in this article, which will allow the reviewers and myself to move your article forward towards publication.

Reviewer 1 ·

Basic reporting

The manuscript explains how to take brain images using microCT and process them to get the volumes of six brain regions: olfactory bulbs, telencephalon, diencephalon, midbrain, cerebellum and brain stem. As far as I can tell, the science presented in the manuscript is basically sound. However, the manuscript is very poorly written, to the point where in places it is difficult to understand what it is attempting to communicate. I recommend the authors undertake several rounds of thorough editing and restructuring to free the manuscript from grammatical, factual, and citation errors, as well as to dramatically improve the clarity and flow.

Some specific comments are included below. I did not examine the supplementary material.

Lines 36-37: “However, detecting individual variation in brain morphology can be difficult, especially in the endotherms (van Schaik Carel P et al., 2021).” The citation is a paper discussing a novel way to control for body size in brain size analyses in primates. It is not clear to me how it supports the preceding sentence. I did not thoroughly check that all the references are appropriate; I recommend the authors go through their references carefully and make sure each citation is appropriate.

Lines 37-38: “In contrast, ectotherms with continuous adult neurogenesis allow for greater neural plasticity and enable the capture of small-scale variation.” I think it is a misconception of the authors that continuous adult neurogenesis and the associated plasticity would have any effect on the ability to capture small-scale variation.

Lines 59-62: “For instance, using imagery methods, such as magnetic resonance imagery (MRI) and X-ray micro-tomography (microCT) to obtain 3D brain morphometric information for brain morphology enhances data quality and makes it more likely to detect small-scale inter individual variation.” I’m not sure how 3D imaging techniques, erroneously referred to here as “imagery methods”, “enhances data quality” and/or “makes it more likely to detect small-scale… variation”. More likely, compared to what? Compared to traditional histological techniques I would argue that 3D imaging is worse for both these things, depending on what exactly is meant by “data quality”.

Lines 66-67: “To our knowledge, there is only one study on the lizard brain, the tawny dragon (Ctenophorus decresii), which has used the MRI technique (Hoops D et al., 2021).” There is also Foss KD, Keller KA, Kehoe SP and Sutton BP (2022) Establishing an MRI-Based Protocol and Atlas of the Bearded Dragon (Pogona vitticeps) Brain.Front. Vet. Sci. 9:886333. doi: 10.3389/fvets.2022.886333

Lines 67-69: “While the study provides a reference brain atlas for the species, it employs manual segmentation that is challenging to reproduce.” The cited paper uses an automated registration technique called MAGeT Brain (Multiple Automatically Generated Templates of different Brains). See the first paragraph of the “Statistical analysis” section of the Methods in the cited paper.

Lines 92-93: “The animals were first used in another study to test their behaviour and cognitive abilities.” Has this study been published? If so, it should be cited here.

Lines 113-114: “During the staining, we were able to check the staining procedure several times within a few weeks until the whole brain was stained.” How was staining checked?

The “Results and Discussion” section is jumbled and difficult to follow. It needs to be reorganized such that it is clear and easy to understand. Nonetheless, the actual results presented seem to be sound.

It appears the “Results and Discussion” section contains very little, if any, discussion. A discussion section, perhaps including elaborations of some of the ideas stated in the Conclusion, would be appreciated. However, the “Results and Discussion” section does appear to include some methods, which should be moved to the Methods section.

Certain results should be elaborated on both in words and in statistics, such as the finding included, essentially in passing, on lines 224-225.

The manuscript would benefit from a few more rounds of editing for flow and clarity. Currently, the manuscript reads in quite a stilted fashion, with many sentences seeming disconnected from those before and after. A few examples:
Lines 41-46: “Lizards can show complex cognitive abilities, such as flexibility and problem-solving, and possess a rich behavioural repertoire. Their geographic widespread distribution and occupancy of a wide range of ecological niches, in addition to their plastic brains, make them ideal for addressing questions about how their brain morphology shapes behaviour and cognition.”
Lines 101-103: “The staining protocol was adapted from to our samples for optimised scans. The samples have been already fixed in 4% PFA and rinsed and stored in PBS.” In this case, in addition to being incongruous, the second sentence is repeating information presented in lines 95-98.

There are grammatical errors throughout. Here is a sample of the grammatical errors I noticed:
Lines 76-77 “…we segmented anatomically six distinct major brain regions: olfactory bulbs, telencephalon, diencephalon, midbrain, cerebellum and brain stem”. This should begin “…we anatomically segmented”.
Line 87: “Animal study was approved under ethics permit…”. This should not start with “Animal” but rather “This” or “Our” or something similar.
Lines 102-103: “The samples have been already fixed in 4% PFA and rinsed and stored in PBS.” “have” should be “had”.
Lines 114-115: “For practicality, this step did not need to take the samples out of the PTA staining solution.” “The step” is not the actor here and should not be the subject of this sentence.

Experimental design

The experimental design, broadly, seems sound, although I do not have experience in AI or "deep learning". The methods section needs several rounds of thorough revision in order to be clear, understandable, and replicable.

Validity of the findings

To the extent that I could understand the results, they appear to be sound.

Reviewer 2 ·

Basic reporting

The work by Zhou et al applied microCT to image 29 lizard brains and tested the performance of two deep-learning segmentation methods using a small number of training samples in the parcellation of the six major brain regions and the associated volumetric estimates. The authors claimed that given as few as 5 datasets for training; this semi-automatic segmentation protocol enables to delivery of satisfying results in an efficient and user-friendly way. This study also suggested that this method would be practical for diverse neurobiological studies, particularly useful for those focused on wild animals with small sample sizes. However, several distinct shortcomings are noted in the current manuscript, as detailed in the attached file.

Experimental design

see detail in the attached file

Validity of the findings

see detail in the attached file

Annotated reviews are not available for download in order to protect the identity of reviewers who chose to remain anonymous.

Reviewer 3 ·

Basic reporting

no comment

Experimental design

no comment

Validity of the findings

no comment

Additional comments

Manuscript #107093

The authors of the article “Brain virtual histology and volume measurement of a lizard species (Podarcis bocagei) using X-ray micro-tomography and deep-learning segmentation” combine X-ray micro-tomography and deep-learning segmentation to measure the volumes of six brain regions in 29 wild lizards. They provide a detailed protocol covering all steps, including sample preparation, and demonstrate that just five data sets are sufficient to train a neural network capable of accurate lizard brain segmentation. Their work is published as open-access, freely available to the scientific community to enhance reproducibility, transparency, and to encourage follow-up studies.
The article explores the potential of deep learning methods to examine links between inter-individual variation in animal behaviour and cognition and brain morphology and function on a large scale, enabling detection of subtle differences. This study is a valuable resource for the biological community, offering a detailed protocol and a critical evaluation of two accessible deep-learning tools for automated segmentation, Biomedisa and AIMOS. The writing is clear and accessible, and the arguments are based on the latest research. However, a few revisions are needed before publication.

Line 59: It should be “imaging methods” and “magnetic resonance imaging”.

Line 150: […] with skip connection.

Line 137: The more common term is “semi-automatically”.

Line 139: Random walk interpolation in Biomedisa.

Line 162: It should be “TensorFlow”.

Line 165: The default parameters of Biomedisa were changed in 2023 due to the evaluation conducted in the supplement of https://doi.org/10.1371/journal.pcbi.1011529. The default network size was adjusted to 32-64-128-256-512, and the number of epochs was set to 100. Although the network size can still be modified in the settings, it would be helpful to clarify which network size was used in this study. One way to identify this is by file size: The older, larger network is about 755.1 MB, whereas the newer, smaller network is around 188.7 MB.

Lines 173-177 and 234ff: The most critical issue for me is that I don’t fully understand the evaluation: For AIMOS, the data was divided into training, validation, and test sets. Although a validation set is technically not required for Biomedisa, it’s strongly recommended—not only for hyper-parameter optimisation (which isn’t relevant here as you use the default setup) but also to ensure that only the best-performing network is saved. Without a validation set, fluctuations during training could result in the final network performing worse than earlier iterations. To address this, the online version of Biomedisa allows specification of a dedicated validation set rather than just a validation split. I recommend using the same training, validation, and test sets for both Biomedisa and AIMOS. To speed up experiments: while using six runs is appreciated, it may not be necessary, as the variance is typically minimal when a validation set is used.

Lines 234ff and Fig. 8: I am unclear on what “All three planes, coronal, sagittal and horizontal, were compared using Biomedisa” means. For AIMOS, this makes sense, as AIMOS is a 2D approach, so the orientation of 2D slices might impact the results. However, since Biomedisa operates in 3D, there shouldn’t be a difference between planes. Does this imply that all datasets were reoriented to observe the effects from different views?

Line 181 & 190: The use of both T and P in equations (1) and (2) could be confusing. If I understand correctly, in equation (2), P and T are scalars representing volume size, calculated as the number of voxels multiplied by voxel size. Consider using V_P and V_T here instead. In equation (1), avoid referring to these as volumes, as they represent sets of voxels in the manually segmented ground truth and predicted segmentations.

Line 202: I think it should be Fig. 3 and 4.

Line 204: The technique is called magnetic resonance imaging.

Line 225: Although I’m not a biologist, I feel a conclusion on the observed variance in brain areas is missing. With a sample size of 29 lizards, is it large enough to support meaningful conclusions? If so, does the 14-19% variance observed in brain areas support the variations in behaviour and cognition observed in lizards?

Line 255: On what basis did the authors determine that accuracy was sufficient with five training datasets? From Fig. 8, it appears that accuracy stabilises after five datasets—was this the basis for their conclusion, or was it assessed visually?

Line 273: Biomedisa offers some data augmentation techniques. Given that “flip along the z-axis” could effectively augment the dataset by mirroring the left-right orientation without changing anatomical structure, was this technique tested, and did it improve results?

Line 367: This reference appears to be a duplicate of the previous one.

Data availability statement: The statement currently mentions only the label data, although the image data is also available on Figshare. However, I was unable to locate a trained network.

Figure 1: Although informative, the image quality is quite low.
Figure 2: Labels (a), (b), and (c) are small and not properly aligned.
Figures 7 & 8: The quality of these figures could be improved by using vector graphics (e.g., SVG). Programs like R or Matplotlib typically allow direct export as SVG instead of PNG.

---

## Round 0.2 · Major Revisions

Dear authors, I ask you to respond very carefully to each point of the reviewers' comments and hope that the new version of this manuscript can be approved for publication.

Reviewer 1 ·

Basic reporting

See attachment

Experimental design

See attachment

Validity of the findings

See attachment

Additional comments

See attachment

Annotated reviews are not available for download in order to protect the identity of reviewers who chose to remain anonymous.

Reviewer 2 ·

Basic reporting

The revised manuscript has been significantly improved and now meets the standards for publication.

Experimental design

The revised manuscript has addressed the shortcomings raised by the reviewer. The current version is easy to understand and follow the experimental steps.

Validity of the findings

This revision provides a clear guideline to demonstrate modern brain imaging techniques on the non-model animal samples. This methodology-focused paper would provide benefits for the groups working with those non-model animals.

Reviewer 3 ·

Basic reporting

no comment

Experimental design

no comment

Validity of the findings

no comment

Additional comments

Thank you for your clarifications and the revised manuscript which reads much easier now. Overall, I believe this is a valid study of interest to anyone conducting similar research. I have just two minor points and one major issue. The line numbers refer to the version with tracked changes:

Minor:
Line 247: My apologies! U-Net has several skip connections. I forgot the „s“ in my original response. You already use the plural form in the rest of the text.

The mathematically correct term in line 279 & 280 is „T is the set of voxels“ and in the formula, you must use the magnitude of the sets, denoted as |T|. Thus, it must be DSC=2*|T ⋂ P| / |T| + |P|. I think I now understand why you initially used volumes, but ⋂ is an operator on sets, not scalars.

Major:
Biomedisa cannot be trained on individual slices, only on complete volumes. Therefore, the statement in lines 369 & 370—“both were trained on the sagittal planes”—is incorrect. Similarly, in lines 326ff (“[…] comparing all three planes, coronal, sagittal, and horizontal using Biomedisa in separate training and testing […]”) and lines 385 & 386 (“[…] all three views, namely coronal, sagittal, and horizontal slices, were compared in training and prediction accuracy […]”), Biomedisa should not be associated with individual plane-based training or prediction.

The only explanation I have is that you validated individual slices from the results and then averaged the Dice scores across slices. While this may introduce slight variations, Dice scores for Biomedisa should be computed in 3D after fully segmenting the volume.

For AIMOS, however, it is valid to calculate Dice scores separately for models trained on coronal, sagittal, and horizontal slices—this distinction is indeed interesting. However, the final Dice score of these differently trained models should also be computed in 3D.

If your evaluation method is restricted to 2D calculations, you should explicitly describe the calculation process. In this case, the differences observed for Biomedisa would be purely statistical, arising from the way the Dice score is computed rather than any inherent differences in segmentation.

Regarding Figure 8, AIMOS should have three separate options (coronal, sagittal, and horizontal), whereas Biomedisa should have only one, without specifying any plane orientation. Since you mention that AIMOS performed best on sagittal slices, including numerical results to support this claim would be valuable.

Line 324-326: please correct: “In comparison, Biomedisa operates in 3D and does not have distinct plane orientations.”

---

## Round 0.3 · Minor Revisions

Dear Dr. Zhou, I ask you to make some minor corrections to the manuscript before I can approve it for publication.

Reviewer 1 ·

Basic reporting

See attached PDF

Experimental design

See attached PDF

Validity of the findings

See attached PDF

Additional comments

See attached PDF

Annotated reviews are not available for download in order to protect the identity of reviewers who chose to remain anonymous.

Reviewer 3 ·

Basic reporting

no comment

Experimental design

no comment

Validity of the findings

no comment

Additional comments

I believe my concerns have been adequately addressed. However, there are still a few technical corrections needed before publication:

1. The Figure 8 description has not been updated to reflect recent changes. It still mentions labels (a) and (b). Additionally, the terms "coronal" and "sagittal" are still used in relation to Biomedisa, which should be revised.

2. The reference Dragunova, Y. 2023 has been removed but is still cited in places, such as in Figure 8 and Table 2.

3. In the reference "Philipp L, Vincent H, Enhancing a diffusion algorithm for 4D image segmentation using local information, ProcSPIE, 2016, pp. 97842L", the first names are written out while the surnames are abbreviated. This is inconsistent with other references, where the first names are abbreviated, and surnames written out.

---

## Round 0.4 · Minor Revisions

Dear Dr. Zhou, I ask you to improve the manuscript in accordance with the reviewer's comments.

Reviewer 1 ·

Basic reporting

Basic reporting is excellent and meets or exceeds all the journal's standards.

Experimental design

Experimental design appears excellent, noting that I am not familiar with AI and cannot comment on that aspect of the design.

Validity of the findings

Findings appear well supported and the underlying data is publicly available. Conclusions are well stated and appropriate in scope.

Additional comments

All my previous comments were addressed and I think this paper is just about ready for publication. I have only two minor comments that I would appreciate it if the authors could address.

Line 134. What is the storage solution?

Line 244-245. That the sagittal plane produces the best segmentation is a result and should be moved back to the results. What I mentioned in my last review was that the hypothesis as to why the sagittal plane works best (because it contains the most brain regions simultaneously) belongs in the discussion. The result should not have been moved to the discussion.

Reviewer 3 ·

Basic reporting

no comment

Experimental design

no comment

Validity of the findings

no comment

Additional comments

All remaining technical points were addressed.

---

## Round 0.5 · accepted · Accept

Dear Dr. Zhou, I am pleased to inform you that this article has been accepted for publication.